# Clinical and Histopathological Effects of Intracameral Ranibizumab in Experimental Trabeculectomy

**DOI:** 10.3390/ijms24087372

**Published:** 2023-04-17

**Authors:** Yaakub Azhany, Wan Faiziah Wan Abdul Rahman, Hasnan Jaafar, Jen Hou Low, Wan Nazirah Wan Yusuf, Ahmad-Tajudin Liza-Sharmini, Jemaima Che Hamzah

**Affiliations:** 1Department of Ophthalmology, Faculty of Medicine, Universiti Kebangsaan Malaysia, Kuala Lumpur 50300, Wilayah Persekutuan, Malaysia; 2Department of Ophthalmology & Visual Science, School of Medical Sciences, Universiti Sains Malaysia, Kubang Kerian 16150, Kelantan, Malaysia; 3Department of Pathology, School of Medical Sciences, Universiti Sains Malaysia, Kubang Kerian 16150, Kelantan, Malaysia; 4Department of Pharmacology, School of Medical Sciences, Universiti Sains Malaysia, Kubang Kerian 16150, Kelantan, Malaysia

**Keywords:** modulation agent, wound healing, ocular surgery, ranibizumab, trabeculectomy

## Abstract

Post-surgical scarring is a known cause of trabeculectomy failure. The aim of this study was to investigate the effectiveness of ranibizumab as an adjuvant anti-scarring agent in experimental trabeculectomy. Forty New Zealand white rabbits were randomised into four eye treatment groups: groups A (control), B (ranibizumab 0.5 mg/mL), C (mitomycin C [MMC] 0.4 mg/mL), and D (ranibizumab 0.5 mg/mL and MMC 0.4 mg/mL). Modified trabeculectomy was performed. Clinical parameters were assessed on post-operative days 1, 2, 3, 7, 14, and 21. Twenty rabbits were euthanised on day 7, and the other twenty were euthanised on day 21. Eye tissue samples were obtained from the rabbits and stained with haematoxylin and eosin (H&E). All treatment groups showed a significant difference in IOP reduction compared with group A (*p* < 0.05). Groups C and D showed a significant difference in bleb status on days 7 (*p* = 0.001) and 21 (*p* = 0.002) relative to group A. H&E staining showed significantly low fibrotic activity (*p* < 0.001) in group C on both days and inflammatory cell grade in group B on day 7 (*p* < 0.001). The grade for new vessel formation was significantly low in groups B and D on day 7 (*p* < 0.001) and in group D on day 21 (*p* = 0.007). Ranibizumab plays a role in reducing scarring, and a single application of the ranibizumab–MMC combination showed a moderate wound-modulating effect in the early post-operative phase.

## 1. Introduction

Trabeculectomy is an important filtration surgical procedure used to reduce intraocular pressure (IOP) in the treatment of patients with glaucoma. However, postoperative scarring is the main issue that causes failure of the procedure. Studies have shown that the leading cause of trabeculectomy failure is the excessive growth of tenons capsule fibroblasts [1]. The administration of mitomycin C (MMC) has been considered a gold standard for modulating the healing process and preventing excessive scarring [2]. However, the use of MMC is not without complications. Endophthalmitis, scleral melting, and conjunctiva leaking are among the known complications [3]. Despite treatment with MMC, long-term bleb failure can still occur [3]. Therefore, a new and safer agent, either as a single agent or in combination with MMC, is needed to overcome fibrosis.

Ranibizumab (Lucentis ranibizumab injection, Genentech, Inc., South San Francisco, CA, USA) is a recombinant humanised anti-VEGF-A monoclonal fragment antigen-binding (Fab) antibody approved by the Food and Drug Administration for the treatment of neovascular age-related macular degeneration (AMD). It was formulated to bind to all biologically active VEGF-A forms by targeting amino acid residues in the ‘80s loop’, which are required in receptor binding [4]. VEGF-A has been reported to mediate increased vascular permeability; induce angiogenesis, vasculogenesis, and endothelial cell growth; promote cell migration; and inhibit apoptosis. VEGF plays an important role in scar tissue formation [5]. Intravitreal ranibizumab has a good safety profile, is effective as an anti-angiogenic mediator in AMD [6], and has shown a possible synergistic action with MMC in trabeculectomy, which results in a more diffuse bleb with less vascularity [7]. The aim of this trial was to compare clinical and histopathological changes in eye tissue samples obtained from New Zealand white rabbits treated with ranibizumab and MMC monotherapies and a combination of both during modified trabeculectomy surgery.

## 2. Results

### 2.1. Clinical Parameters

The status of the experimental and control eyes was evaluated clinically on designated days. No cases of endophthalmitis were observed. The rabbits in all the groups showed no signs of cataracts during the study period. IOP measurements were significantly lower in groups C and D than in the control group (*p* = 0.003 and 0.013, respectively) on all measurement days (Figure 1, Table 1). Bleb status was significantly good on post-operative days 7 and 21 in groups C and D (*p* = 0.001 and 0.002, respectively) (Figure 2, Table 2). Conjunctival inflammation was significantly lower on post-operative day 7 in group C (*p* = 0.008). Anterior chamber activity was significantly lower on post-operative day 7 in groups B and D (*p* = 0.030). The anterior chamber depth was well formed in all the groups, with no statistically significant differences between the groups. Detailed comparisons of the clinical results between the groups are shown in Table 2. Figure 2 shows the clinical observations according to the groups.

### 2.2. Histopathological Findings

The histopathological analysis of the specimens was performed from the centre of the sclerotomy site through the centre of the cannula. A comparison of histopathological grades between the groups is shown in Table 3. Fibrotic activity was significantly lower in group C on days 7 and 21 (*p* < 0.001 and 0.009, respectively). The inhibition of blood vessel formation was statistically significant on post-operative day 7 (*p* < 0.001) in groups B and D, and on day 21 in group D (*p* = 0.007). The grading of interstitial oedema was significantly lower in groups B and D on day 7. The grading of inflammatory cells was significantly lower in groups B and C on day 7 and in group D on day 21 (*p* < 0.001 and 0.004, respectively).

The histopathological examination findings of all four groups at different magnifications and time frames are shown in Figure 3, Figure 4 and Figure 5.

Generally, the MMC group shows the most obvious lower tissue infiltration into the cannula lumen at both times, while MMC–Rani group shows the most prominent lower infiltration at post op day 7 but not at day 21. (Figure 3). These are supported by predominantly loose collagen fibres with loose subconjunctival stroma and the fibroblasts are reduced in density and have fewer inflammatory infiltrates in MMC group (Figure 4C) and MMC-Rani group (Figure 4D) at post-op day 7. While at post-op day 21, as compared to control group, all other groups show a strong cellular response due to the increased proliferation of fibroblasts, producing collagen fibres, increased in capillary blood vessels (Figure 5B–D).

## 3. Discussion

The success of filtrating surgery is based on IOP reduction, and bleb morphology indirectly justifies the function of the surgery. This study shows that the inhibition of VEGF expression using ranibizumab affects the acute phase of wound healing, with significant histological differences in the number of inflammatory cells, fibrosis activity, and interstitial oedema, in addition to its main role in curbing new vessel formation. However, these effects of ranibizumab were not pronounced in the late phase, on day 21. This may indicate that a repeat dose is necessary to produce a more sustained effect, possibly through repeated topical administration [8]. This may be due to the shorter half-life of ranibizumab (2.8 days in vitreous) relative to other anti-VEGF agents such as bevacizumab (4.32 days) [9]. The effect of MMC predominated clinically when these two agents were combined. The impact of the combination was lower than that of MMC alone. This finding gives the impression that ranibizumab has the potential to be an additional agent for situations that require a moderate modulatory effect when the effects can be sustained for oedema and inflammatory cells.

The main advantage of these observations was that ranibizumab significantly showed the least anterior chamber reaction in the acute phase. In terms of IOP, the results of the single or combined ranibizumab groups were equivalent, whether in the acute or late phase, but better than those of the control group. No significant difference in IOP was expected between the groups, as high IOP was not induced before treatment. However, if the bleb was functioning and fibrosis was retarded, the IOP was expected to be indirectly reduced even in a non-glaucoma model. In this study, the IOP reduction was not consistent with the bleb morphology clinically or histopathologically. Histopathological observation revealed that the main advantage of administering ranibizumab 0.5 mg/mL was the inhibition of the formation of new blood vessels and the control of inflammatory cell growth. In the late phase, MMC dominated the effect in all observations.

The most significant effect of ranibizumab 0.5 mg/mL was observed in the acute phase, on day 7, but decreased with time. This leads to the question ‘Should the use of ranibizumab 0.5 mg/mL in glaucoma surgery be given periodically?’ The effect of ranibizumab in controlling macular oedema secondary to retinal vein occlusion, for example, requires periodic intravitreal injections; that is, three consecutive injections within a month before re-evaluation [10]. The intravitreal injection of other anti-VEGF agents such as bevacizumab has been found to play a role in iris neo vessel regression and IOP control in cases of neovascular glaucoma (NVG), and potentially affects the success rate of trabeculectomy with MMC administration. However, this effect lasts for a limited time, and re-injection was recommended to maintain this effect [11].

In their study, Kahook et al. used intravitreal ranibizumab 0.5 mg/mL during and, if required, in the first month after surgery [7]. A previous study also reported this observation, where the early post-surgical outcomes were highly encouraging, with less vascularity and better blebs than in MMC. However, this positive effect was attenuated, and long-term effects were found to show a higher vascular effect of IOP elevation when this agent was used alone [12,13,14].

The observations from this experiment showed that the effects of ranibizumab were short-term in nature. This study answers the question posed by the observation of case series and case reports that reported the use of ranibizumab in trabeculectomy for patients with NVG, where long-term success was not significant [15]. However, a previous study demonstrated that the benefit of intravitreal anti-VEGF injections was that the incidence of complications during surgery, such as hyphaema, was reduced [16].

Several studies have shown the positive short- and long-term effects of various regimens and types of anti-VEGF agents in cases of NVG. However, the primary cause of NVG must be addressed to stabilise the disease [17,18,19,20,21]. The long-term success of trabeculectomy with a concurrent use of bevacizumab injection in NVG cases is not as convincing as that with MMC. However, positive changes in bleb vascularity have been reported. Similar observations were also reported with the use of glaucoma drainage devices in NVG [22].

This study has several limitations that can be improved on future studies. Its duration was only 21 days. A more extended period, such as 1, 2, or 3 months, allows for observations of the trend of the healing process under the influence of ranibizumab and MMC modulating agents. The long-term effectiveness of the combination of ranibizumab and MMC can also be observed. Although the present study elucidated that the action of ranibizumab is focused within the acute phase and is not effective in the late phase of 21 days, a longer observation of its efficacy after 3 or 6 months should be ascertained with single or repeated subconjunctival injections.

We suggest that future studies include immunohistochemical tests on animal tissue sections. This technique can determine the presence and level of specific cellular proteins (antibodies) involved, such as endothelial cells, macrophages, platelets, and Ki-67. Special staining is recommended to determine the proteins that explain any observation. In addition, the method of ranibizumab treatment on rabbits can also be modified, such as repeating the dose after surgery at certain intervals/periods or continuing with ranibizumab eye drops post-operatively. In addition, whether or not the effectiveness of the treatment increases with this modification must be observed.

## 4. Materials and Methods

This was an experimental study. Forty New Zealand white rabbits were randomised into four groups: group A, the controls (*n* = 10); group B, those that received intracameral ranibizumab 0.5 mg/mL (*n* = 10); group C, those that received MMC 0.04% (*n* = 10); and group D, the combination group (intracameral ranibizumab 0.5 mg/mL and MMC 0.04%; *n* = 10). The study was conducted at the Animal Research and Service Centre, University Sains Malaysia.

### 4.1. Animals

All animal procedures and methods for securing the animal tissue samples complied with the Association for Research in Vision and Ophthalmology Statement for the Use of Animals in Ophthalmic and Vision Research and our institutional guidelines. Forty New Zealand white rabbits aged between 12 and 14 weeks and weighing 2.0 to 2.5 kg were used. Prior to the experiments, all rabbits underwent a 1-week acclimatisation period.

### 4.2. Anaesthesia

The induction of general anaesthesia was started with an intramuscular injection of ketamine (100 mg/mL; Vetoquinol UK Limited, Towcester, UK) at a dose of 0.35 mg/kg and xylazine (20 mg/mL; Indian Immunologicals Limited, Hyderabad, India) at a dose of 5 mg/kg. This was followed by the administration of isoflurane gas (inhalation liquid 100%; Piramal Healthcare Limited, Mumbai, India), which was 1–3% at the start of surgery and 3% for the rest of surgery. The rabbit’s eyes were then instilled with a drop of temporary anaesthetic (proparacaine hydrochloride; Alcaine, Pharmaco [NZ] Limited, Auckland, New Zealand). The rabbit’s pulse rate and blood pressure were monitored throughout the surgery. At the end of the surgery, the rabbit was injected intramuscularly with meloxicam 0.3–0.6 mg/kg (5 mg/mL; Intas Pharmaceuticals Limited, Ahmedabad, India) as a post-surgical pain reliever.

### 4.3. Technique of Surgery

A lid speculum was inserted to expose the bulbar conjunctiva, and a drop of iodine was instilled and washed out with a balanced salt solution (Alcon Laboratories, Fort Worth, TX, USA). Two partial-thickness 6–0 silk corneal traction sutures were placed superiorly, and the eye was pulled inferiorly. A fornix-based conjunctival flap was fashioned, measuring 5 mm in length at limbus, followed by a 3 × 4 mm limbus-based rectangular scleral flap outlined with a triangular steel blade. A partial-thickness scleral flap was then dissected carefully, starting at 2 mm behind the limbus and continuing until the blade was visible in the anterior cornea stroma. For groups C and D, 4 thin sponges of 0.2 × 0.2 mm in size were moistened with MMC 0.4 mg. This MMC-moistened sponge was then placed on the scleral surface around the scleral flap area but under the conjunctival flap for 3 min. Then, the eye was washed with 60 mL of saline water.

A modified trabeculectomy technique [23] using a cannula to maintain a patent scleral tract was used. It was performed by inserting a 22 g, 25 mm Venfon intravenous cannula (BD Biosciences, Franklin Lakes, NJ, USA; Figure 6). This cannula was inserted into the anterior chamber of the eye until it approached the border of the pupil, after which the needle was withdrawn, and the plastic cannula was left in situ. The cannula was cut approximately 1 mm from the entry point. This plastic cannula was sutured using 10-0 Ethilon thread. The scleral flap was sutured with Ethilon 10-0, and some aqueous was allowed to seep out. The conjunctiva was then sutured with Ethilon 10-0 using the ‘purse’ bonding technique. For groups B and D, as an additional step, ranibizumab 0.5 mg/mL was intracamerally injected into the anterior chamber diagonally through the corneal wall at a 45° angle to avoid leakage on the cornea.

The Tenon’s capsule was closed with interrupted sutures, and a conjunctival incision was closed separately with a 10-0 nylon suture (Alcon, Fort Worth, TX, USA) for a water-tight closure. All glaucoma filtration surgeries were performed by a single surgeon with experience operating on rabbit eyes. All rabbits underwent filtration surgery on the right eye only and were randomly allocated to one of four treatment groups. At the end of the operation, the rabbits received subconjunctival dexamethasone and gentamycin injections. Levofloxacin 0.5% eye drops (Cravit, Santen) were administered, and the eyes were padded for 24 h. Subsequently, the administration of levofloxacin eye drops TDS was continued for 3 days.

### 4.4. Clinical Evaluation

Postoperative IOP was measured in both eyes using an applanation tonometer (Tono-Pen Avia, Reichert, Inc., Depew, NY, USA) under topical anaesthesia at 7:00 to 8:00 AM on days 1, 7, 14, and 21. The mean reading of three measurements was recorded. A second investigator who recorded the IOP was blinded to the groups. A slit lamp examination was performed to examine the anterior chamber depth, bleb status, conjunctiva inflammation, and anterior chamber activity.

Anterior chamber depth was recorded as shallow (1), moderate (2), or deep (3). Bleb status was recorded as not visible (0), slightly visible (1), and clearly visible (2). Conjunctiva inflammation was recorded as slight + (1), moderate ++ (2), and severe +++ (3). Anterior chamber activity was recorded as ‘fibrin was not visible + (1)’, ‘fibrin was slightly visible at the tip of the plastic cannula tube ++ (2)’, and ‘fibrin was seen abundantly in or along the plastic cannula tube +++ (3)’.

### 4.5. Histopathological Evaluation

On postoperative days 7 and 21, all animals (20 animals each time) were euthanised with a lethal intravenous injection of excess phenobarbitone, and the tissues were processed for histopathology. Both eyes were enucleated. The upper lid was left intact and attached to the globe to preserve the architecture of the superior fornix and conjunctival tissues around the drainage site. All eyes were fixed in formalin acetic acid alcohol solution for 24 h, stored in 70% alcohol, and fixed in paraffin wax. Sequential 4 µm sections of the operative wound site across the cannula tube were prepared and stained with haematoxylin and eosin to obtain a general impression of total cellularity.

### 4.6. Statistical Analysis

Statistical analysis was performed to determine the differences in IOP, bleb status, conjunctival inflammation, anterior chamber activity, and depth among the four groups using analysis of variance (ANOVA; SPSS 25.0 software, SPSS Inc., Chicago, IL, USA). For multiple comparisons of IOP differences (overall) between multiple pairs of the treatment groups over multiple times, two-way repeated-measures ANOVA was used, and *p* < 0.05 was considered significant. The Fisher exact test was performed for other analyses. As for the histopathologic findings, the differences in all parameters were analysed using the Fisher exact test.

### 4.7. Animal Ethical Approval

The experiments were conducted in accordance with the Association for Research in Vision and Ophthalmology Statement for the Use of Animals in Ophthalmic and Vision Research. Approval from the Animal Ethics Committee of Universiti Sains Malaysia [Animal Ethics Approval: USM/Animal Ethics Approval/2012/(81)(428)] was obtained for this study.

## 5. Conclusions

In this study, the inhibitory activity of the MMC-ranibizumab combination showed a moderate effect which was better than that of single-agent ranibizumab but less effective than single-agent MMC, and it was effective in the acute phase of wound healing.

## Figures and Tables

**Figure 1 ijms-24-07372-f001:**
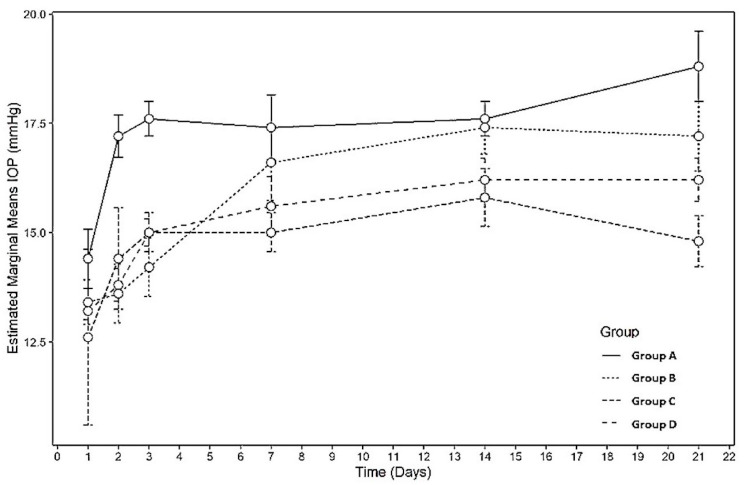
Profile plot showing different IOP readings between difference treatment groups at days 1, 2, 3, 7, 14, and 21. (*n* = 40).

**Figure 2 ijms-24-07372-f002:**
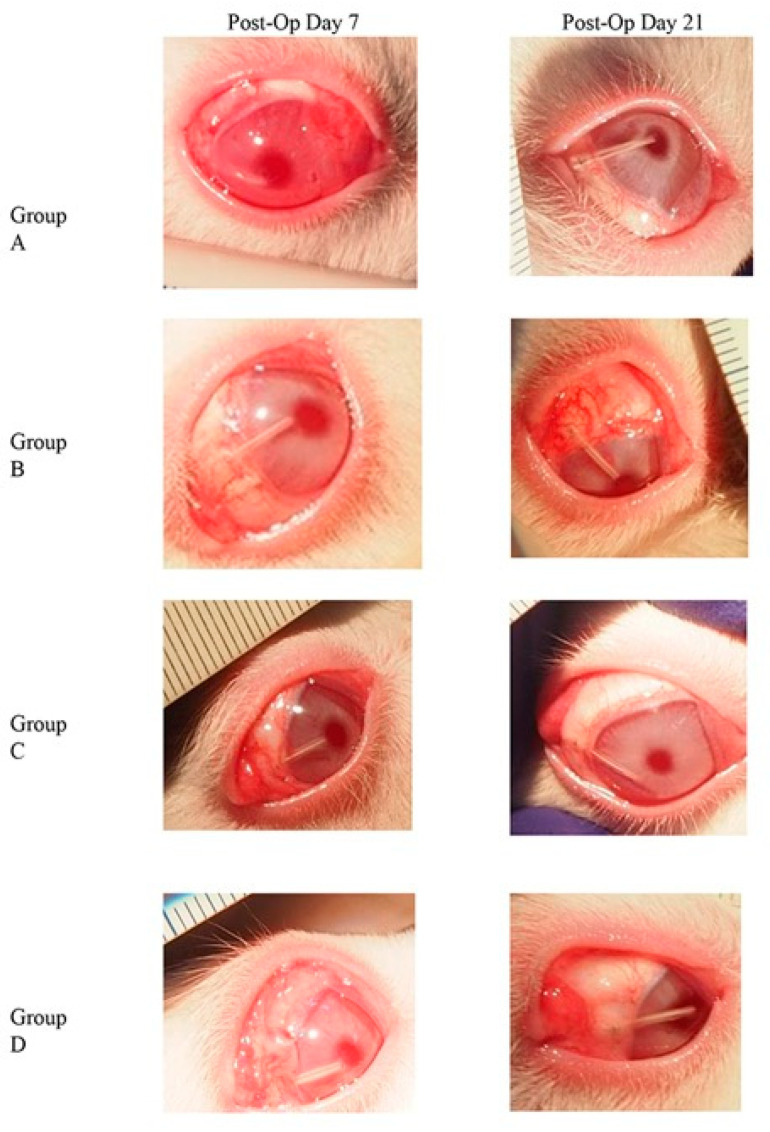
Clinical observations on day 7 and day 21 post trabeculectomy. Through out group A to D, groups C and D showed good status of bleb on post-operative days 7 and 21. Conjunctival inflammation was less seen on post-operative day 7 in group C.

**Figure 3 ijms-24-07372-f003:**
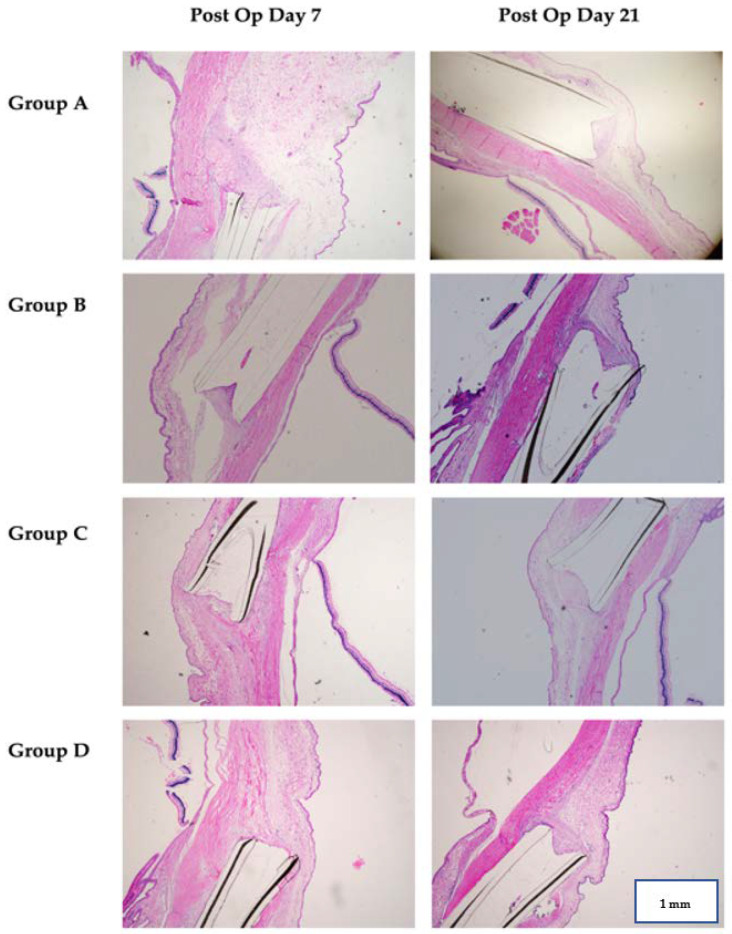
Histopathological findings of all four groups on post operative day 7 and day 21. A cut was made at the centre of the cannula lumen. Group A (control group) shows a low tissue infiltration into the cannula at day 7 but higher at day 21. Group B (Ranibizumab group) shows higher tissue infiltration at both day 7 and day 21. Group C (MMC group) shows a low tissue infiltration into the cannula at both day 7 and day 21. Group D (MMC-Ranibizumab group) shows no tissue infiltration at day 7 but higher at day 21. Generally, the MMC group shows the most obvious lower tissue infiltration into the cannula lumen at both times, while MMC–Rani group shows the most prominent lower infiltration at post op day 7 but not at day 21. (H&E staining, 40× magnification).

**Figure 4 ijms-24-07372-f004:**
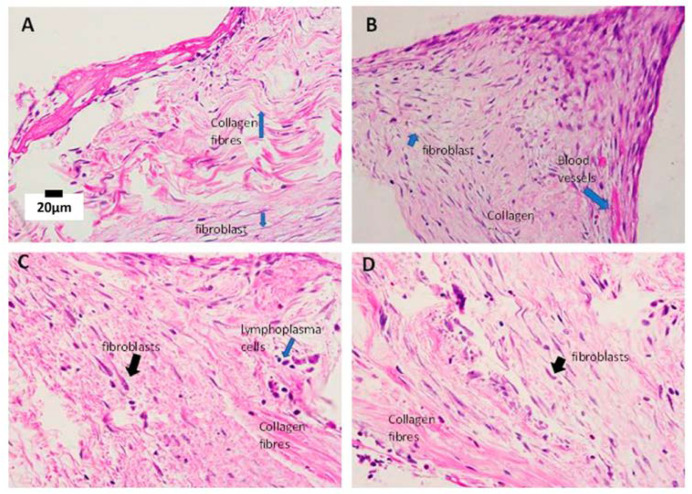
Histopathological findings in all study groups at post-op day 7. (**A**) Group A shows a bundle of densely packed collagen fibres interspersed between the fibroblasts, indicating a cellular response. (**B**) Group B, with haphazardly arranged collagen fibres with increased density of fibroblasts. (**C**) Group C, with predominantly loose collagen fibres (pink eosinophilic stain) with loose subconjunctival stroma. The fibroblasts are reduced in density and have fewer inflammatory infiltrates. (**D**) Group D shows a reduced number of fibroblasts. Dense collagen fibres are noted in the area near the lumen cannula with a cluster of inflammatory cells. Blue and black arrows indicate the labelling of the specific cells. (H&E staining, 400× magnification).

**Figure 5 ijms-24-07372-f005:**
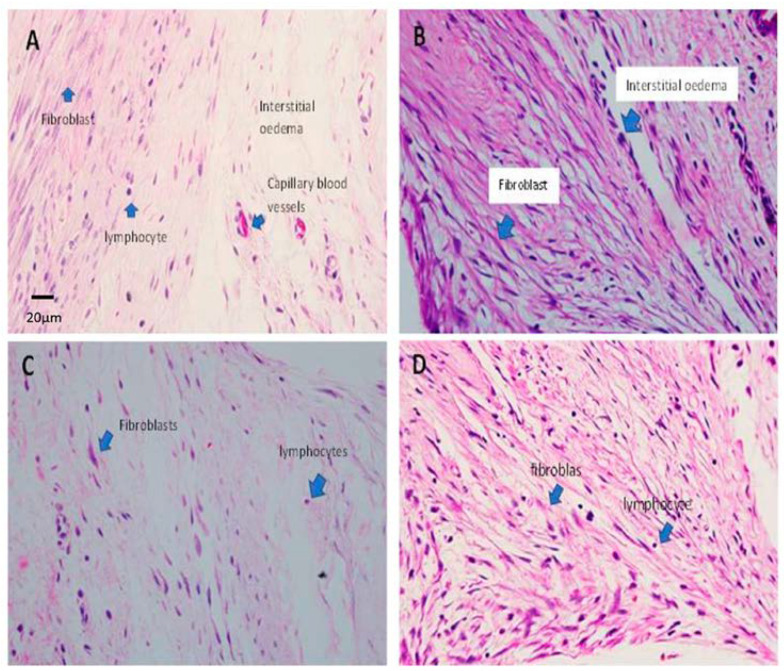
Histopathological findings of all study groups at post-op day 21. (**A**) Group A shows dense fibroblasts, which are characterised by spindled-shaped cells that produce collagen, forming scar tissue. The lymphocyte infiltrates are scattered within the tissue. The interstitial oedema is obvious. (**B**) Group B shows a strong cellular response due to the increased proliferation of fibroblasts (bluish spindled nuclei) and collagen fibres (pink eosinophilic stain fibres). Note the increase in the number of capillary blood vessels lined by endothelial cells. (**C**) Group C shows an increased cellular response with a proliferation of fibroblasts. The fibroblasts are densely packed and plumped, producing collagen fibres. (**D**) Group D shows an increased cellular response with a proliferation of fibroblasts. The fibroblasts are densely packed and plumped, producing collagen fibres. Lymphocytes are scattered but reduce in number. Blue arrows in all images pointing the labelling structures (H&E staining, 400× magnification).

**Figure 6 ijms-24-07372-f006:**
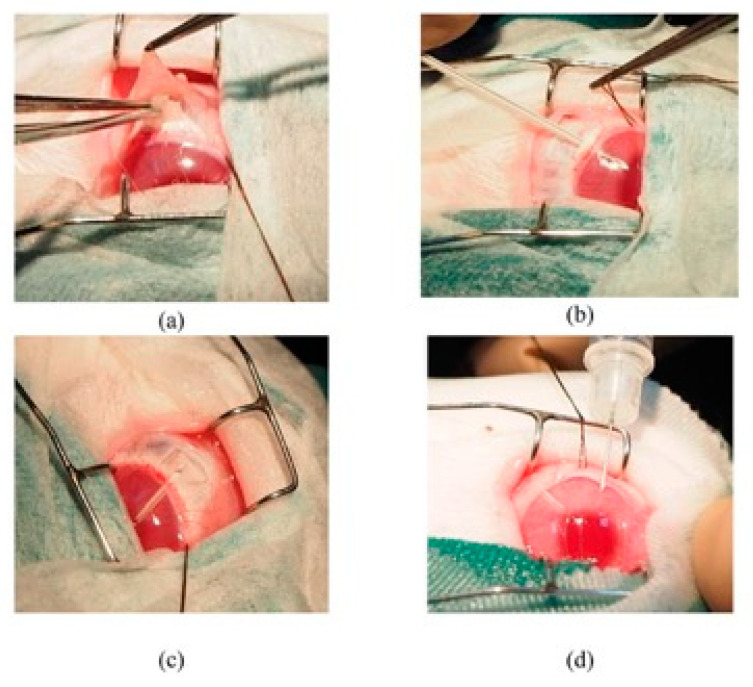
Photos (**a**–**d**) showing the important surgical steps in this experimental trabeculectomy.

**Table 1 ijms-24-07372-t001:** Multiple comparison of IOP by different treatment groups using RM ANOVA.

Rabbits’ Eyes, *n* = 40		
Comparison (Day 1, 2, 3 and 7)	Mean Difference (95% CI)	*p*-value *
Group A vs. Group B	1.60 (0.23, 2.97)	**0.015**
Group A vs. Group C	1.55 (0.18, 2.92)	**0.019**
Group A vs. Group D	1.55 (0.18, 2.92)	**0.019**
Group B vs. Group C	−0.05 (−1.42, 1.32)	>0.999
Group B vs. Group D	−0.05 (−1.42, 1.32)	>0.999
Group C vs. Group D	0.00 (−1.37, 1.37)	>0.999
Rabbits’ eyes, *n* = 20		
Comparison (Day 1, 2, 3, 7, 14 and 21)	Mean Difference (95% CI)	*p*-value *
Group A vs. Group B	1.77 (−0.01, 3.55)	**0.052**
Group A vs. Group C	2.57 (0.79, 4.35)	**0.003**
Group A vs. Group D	2.17 (0.39, 3.95)	**0.013**
Group B vs. Group C	0.80 (−0.98, 2.58)	>0.999
Group B vs. Group D	0.04 (−1.38, 2.18)	>0.999
Group C vs. Group D	−0.40 (−2.18, 1.38)	>0.999

* Adjusted with Bonferonni Correction.

**Table 2 ijms-24-07372-t002:** Comparison of clinical grading among groups on days 7 and 21 postoperatively (*n* = 20).

	Grade *n* (%)	Grade 0 *n* (%)	Grade 1 *n* (%)	Grade 2 *n* (%)	χ^2^-Stat	*p*-Value *
**Bleb Status**						
Day 7	Group A	0 (0.0)	5 (62.5)	0 (0.00)	14.21	**0.001**
	Group B	0 (0.0)	3 (37.5)	2 (16.7)		
	Group C	0 (0.0)	0 (0.0)	**5 (41.7)**		
	Group D	0 (0.0)	0 (0.0)	**5 (41.7)**		
Day 21	Group A	1 (100.0)	3 (33.3)	1 (10.0)	15.20	**0.002**
	Group B	0 (0.0)	5 (55.6)	0 (0.0)		
	Group C	0 (0.0)	1 (11.1)	**4 (40.0)**		
	Group D	0 (0.0)	0 (0.0)	**5 (50.0)**		
**Conjunctiva Inflammation**						
Day 7	Group A	0 (0.0)	5 (55.6)	0 (0.0)	11.47	**0.008**
	Group B	2 (18.2)	3 (33.3)	0 (0.0)		
	Group C	**5 (45.5)**	0 (0.0)	0 (0.0)		
	Group D	**4 (36.4)**	1 (11.1)	0 (0.0)		
Day 21	Group A	2 (15.4)	3 (42.9)	0 (0.0)	7.77	0.063
	Group B	1 (7.7)	4 (57.1)	0 (0.0)		
	Group C	5 (38.5)	0 (0.0)	0 (0.0)		
	Group D	4 (38.5)	1 (0.0)	0 (0.0)		
**Anterior Chamber Activity**						
Day 7	Group A	1 (7.1)	3 (60.0)	1 (100.0)	10.42	**0.030**
	Group B	**5 (35.7)**	0 (0.0)	0 (0.0)		
	Group C	3 (21.4)	2 (40.0)	0 (0.0)		
	Group D	**5 (35.7)**	0 (0.0)	0 (0.0)		
Day 21	Group A	4 (16.67)	1 (100.0)	0 (0.00)	2.96	>0.999
	Group B	5 (27.78)	0 (0.0)	0 (0.00)		
	Group C	5 (27.78)	0 (0.0)	0 (0.00)		
	Group D	5 (27.78)	0 (0.0)	0 (0.00)		
**Anterior Chamber depth**						
Day 7	Group A	0 (0.0)	1 (50.0)	4 (22.2)	2.32	>0.999
	Group B	0 (0.0)	1 (50.0)	4 (22.2)		
	Group C	0 (0.0)	0 (0.0)	5 (27.8)		
	Group D	0 (0.0)	0 (0.0)	5 (27.8)		
Day 21	Group A	0 (0.00)	0 (0.00)	5 (25.0)	-	-
	Group B	0 (0.00)	0 (0.00)	5 (25.0)		
	Group C	0 (0.00)	0 (0.00)	5 (25.0)		
	Group D	0 (0.00)	0 (0.00)	5 (25.0)		

* Fisher’s Exact Test.

**Table 3 ijms-24-07372-t003:** Comparison of histopathological grading on days 7 and 21 post operatively (*n* = 20).

Day	Group	Grading *n* (%)				
		Grade 0 *n* (%)	Grade 1 *n* (%)	Grade 2 *n* (%)	Grade 3 *n* (%)	*p*-Value *
Inflammatory cells						
Day 7	Group A	0 (0.00)	0 (0.00)	3 (37.50)	2 (100.00)	**<0.001**
	Group B	**5 (100.0)**	0 (0.00)	0 (0.00)	0 (0.00)	
	Group C	0 (0.00)	0 (0.00)	5 (100.00)	0 (0.00)	
	Group D	0 (0.00)	5 (100.00)	0 (0.00)	0 (0.00)	
Day 21	Group A	0 (0.00)	5 (50.00)	0 (0.00)	0 (0.00)	**0.004**
	Group B	0 (0.00)	1(10.00)	4 (57.10)	0 (0.00)	
	Group C	0 (0.00)	2 (20.00)	3 (42.90)	0 (0.00)	
	Group D	**3 (100.00)**	2 (20.00)	0 (0.00)	0 (0.00)	
Fibrosis activity						
Day 7	Group A	0 (0.00)	0 (0.00)	5 (100.00)	0 (0.00)	**<0.001**
	Group B	0 (0.00)	5 (71.4)	0 (0.00)	0 (0.00)	
	Group C	**5 (62.5)**	0 (0.00)	0 (0.00)	0 (0.00)	
	Group D	3 (37.5)	2 (28.6)	0 (0.00)	0 (0.00)	
Day 21	Group A	0 (0.00)	0 (0.00)	5 (45.50)	0 (0.00)	**0.009**
	Group B	0 (0.00)	2 (50.00)	3 (27.30)	0 (0.00)	
	Group C	**2 (100.00)**	2 (50.00)	0 (0.00)	1 (33.33)	
	Group D	0 (0.00)	0 (0.00)	3 (27.3)	2 (66.67)	
Interstitial oedema						
Day 7	Group A	0 (0.00)	5 (100.00)	0 (0.00)	0 (0.00)	**<0.001**
	Group B	**5 (50.00)**	0 (0.00)	0 (0.00)	0 (0.00)	
	Group C	0 (0.00)	0 (0.00)	0 (0.00)	5 (100.00)	
	Group D	**5 (50.00)**	0 (0.00)	0 (0.00)	0 (0.00)	
Day 21	Group A	0 (0.00)	0 (0.00)	5 (50.00)	0 (0.00)	0.33
	Group B	1 (50.00)	3 (75.00)	0 (0.00)	1 (25.5)	
	Group C	0 (0.00)	1 (25.00)	2 (50.00)	2 (50.00)	
	Group D	1 (50.00)	0 (0.00)	3 (30.00)	1 (25.00)	
New vessels formation						
Day 7	Group A	0 (0.00)	2(28.60)	3 (100.00)		**<0.001**
	Group B	**5 (50.00**)	0 (0.00)	0 (0.00)		
	Group C	0 (0.00)	5 (71.40)	0 (0.00)		
	Group D	**5 (50.00)**	0 (0.00)	0 (0.00)		
Day 21	Group A	0 (0.00)	5 (35.70)	0 (0.00)		**0.007**
	Group B	0 (0.00)	5 (35.70)	0 (0.00)		
	Group C	1 (20.00)	3 (21.40)	1 (100.00)		
	Group D	**4 (80.00)**	1 (7.10)	0 (0.00)		

* Fisher’s Exact Test.

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
