# Peer review of "Clinical and Histopathological Effects of Intracameral Ranibizumab in Experimental Trabeculectomy"

_ijms, 2023, doi:10.3390/ijms24087372_

Round 1
Reviewer 1 Report
Azhany et al. performed trabeculectomies rabbit eyes to explore the role of anti-VEGF antibodies as an anti-fibrotic adjuvant. Rabbits were randomized to control, ranibizumab, mitomycin-C (MMC), or ranibizumab + MMC. Outcomes were graded based on intraocular pressure, clinical examination of the bleb and eye, and histologic findings.
In my review of table 2, I do not see any benefit to the scores provided by ranibizumab that is not already accounted for by the possible effects of MMC. In other words, there is not a scoring category in which group B provides benefit over group C. The only exception is in Anterior Chamber Activity at Day 7; however, this difference was gone by Day 21. Furthermore, the statistical analysis of Table 2 seems to stem mostly from the differences between the control (Group A) and the rest of the groups, rather than any effects of ranibizumab.
Histologically, ranibizumab seemed to promote fibrosis in the combination group at Day 21. I agree with the authors that there seems to be an early inhibition of neovascularization in the ranibizumab groups that is somewhat lost by Day 21.
It is important to note that in human patients, the purpose of the trabeculectomy is to reduce the intraocular pressure. Bleb characteristics, inflammation, and fibrosis are aspects clinicians consider in order to manage the bleb and maintain IOP reduction. In this paper, a pre-operative IOP would improve the understanding of the effects of MMC or anti-VEGF agents on the postoperative IOP. If there was a sustained intraocular pressure reduction, then further investigation into the bleb characteristics and histology may offer insight as to why the pressure was effectively reduced. Without IOP reduction, the additional analysis does not provide meaningful insight.
1. In the abstract: revise sentence “Group C and D showed significant bleb status differences at D7 and Day-21” so that it makes sense
2. In the introduction, the authors state that there are adverse effects of MMC so that safer agents to overcome fibrosis would be beneficial for trabeculectomies. Their discussion point (line 154 – 155) that combination of the two would help to avoid complications from MMC is somewhat confusing and contradictory.
3. Discussion line 119 – 120: the statement that ranibizumab plays a major role in the acute phase of wound healing is not well substantiated by this paper. First, I think the authors means that VEGF plays a major role and or that inhibition of VEGFa affects the acute phase of wound healing. However their results do not prove this.
4. Line 122-123 is not a complete sentence.
5. Line 143 – 147: The use of reference 10, a case series on neovascular glaucoma does not support the statement that anti-VEGF agents “increase the success rates of trabeculectomy with mitomycin-C”.
6. Recommend grammatical review of the discussion to correct for errors.
Reviewer 2 Report
I had the pleasure to review this manuscript.
This is a very well written paper.
This study should be clarified whether this study is about POAG or NVG.
NVG and POAG have completely different mechanisms of actions.
It is well known that anti-VEGF agents reduce IOP for NVG.
It would be an interesting finding if anti-VEGF agents could have a positive effect, even temporarily, on POAG.
I think the concept of using anti-VEGF agents after trabeculectomy for POAG, when the filtration blebs have collapsed, is a good one.
(I think that you should not mention about NVG in the DISCUSSION.)
Various cytokines are involved to scar the filtration bleb.
You should elaborate more on how the anti-VEGF agents work to inhibit scarring.
Round 2
Reviewer 1 Report
Thank you for addressing the points.
I think the largest drawback of this study is still that the postulated role of anti-VEGF in aiding IOP lowering did not pan out.
The English and grammar still needs to be edited.
Author Response
Dear Reviewer 1,
Thank you for your comment and we do understand your intention. We have already added to the discussion on that matter.
Comment 1:
I think the largest drawback of this study is still that the postulated role of anti-VEGF in aiding IOP lowering did not pan out.
Rebuttal 1 : (line 145-155 in the Discussion section)
In terms of IOP, the results of the single or combined ranibizumab groups were equivalent, whether in the acute or late phase, but better than those of the control group. No significant difference in IOP was expected between the groups, as high IOP was not induced before treatment. However, if the bleb is functioning and fibrosis is retarded, the IOP is expected to be indirectly reduced even in a non-glaucoma model. In this study, the IOP reduction was not consistent with the bleb morphology clinically and histopathologically. Histopathological observation revealed that the main advantage of administering ranibizumab 0.5 mg/ml was the inhibition of the formation of new blood vessels and control of inflammatory cell growth. In the late phase, MMC dominated the effect in all observations.
Comment 2:
The English and grammar still need to be edited.
Rebuttal 2:
We have already got the Profesional English Editing (Scribendi)
Thank You

Reviewer 2 Report
All points raised were corrected.
Author Response
Dear Reviewer 2,
Thank you for your comment and acknowledgment.
Comment 1:
All points raised were corrected.
Rebuttal 1:
Thank you very much for your kind comments
